# Structural Biology Inspired Development of a Series of Human Peroxisome Proliferator-Activated Receptor Gamma (PPARγ) Ligands: From Agonist to Antagonist

**DOI:** 10.3390/ijms24043940

**Published:** 2023-02-15

**Authors:** Hiroyuki Miyachi

**Affiliations:** Lead Exploration Unit, Drug Discovery Initiative, The University of Tokyo, 7-3-1 Hongo, Bunkyo-ku, Tokyo 113-0033, Japan; miyachi_hiroyuki@mol.f.u-tokyo.ac.jp

**Keywords:** peroxisome proliferator-activated receptor gamma, PPARγ, antagonist, structural biology, ligand superfamily concept

## Abstract

Recent progress in the structural and molecular pharmacological understanding of the nuclear receptor, peroxisome proliferator-activated receptor gamma (hPPARγ)—a transcription factor with pleiotropic effects on biological responses—has enabled the investigation of various graded hPPARγ ligands (full agonist, partial agonist, and antagonist). Such ligands are useful tools to investigate the functions of hPPARγ in detail and are also candidate drugs for the treatment of hPPARγ-mediated diseases, such as metabolic syndrome and cancer. This review summarizes our medicinal chemistry research on the design, synthesis, and pharmacological evaluation of a covalent-binding and non-covalent-binding hPPARγ antagonist, both of which have been created based on our working hypothesis of the helix 12 (H12) holding induction/inhibition concept. X-ray crystallographic analyses of our representative antagonists complexed with an hPPARγ ligand binding domain (LBD) indicated the unique binding modes of hPPARγ LBD, which are quite different from the binding modes observed for hPPARγ agonists and partial agonists.

## 1. Peroxisome Proliferator-Activated Receptor Gamma (PPARγ)

Human peroxisome proliferator-activated receptors (hPPARs) are ligand-mediated transcription factors belonging to the human 48 nuclear receptor (NR) superfamily, which includes the retinoid receptor, steroid receptor, vitamin D receptor, and others [1]. Three subtypes of hPPARs—hPPARα [NR1C1], hPPARδ [NR1C2], and hPPARγ [NR1C3]—have been identified to date in various species, including humans [2]. Upon endogenous and/or exogenous agonist binding, hPPARs heterodimerize with another nuclear receptor—the retinoid X receptor—in the nucleus and these heterodimers regulate gene expression by binding to the specific consensus DNA sequences, termed peroxisome proliferator responsive elements (PPREs) in the promoter regions of the target genes. The structural basis of PPREs is a direct repeat of the hexametric AGGTCA recognition motif, separated by one nucleotide (termed DR1) [3].

The hPPARs have five conserved structural domains: A–E domains from the N- to C-terminus [4]. The N-terminal A and B domains contain activation function 1 (AF1), which is involved in ligand-independent coregulator binding. The C domain functions as a DNA binding domain and is the most conserved domain among the hPPARs. The D domain functions as a flexible hinge allowing rotation between the DNA binding domain (C domain) and ligand binding domain (E domain), as well as containing a nuclear localization signal. The E domain is the largest domain in hPPARs and is the second most conserved domain among the hPPARs. There are four main functions of the E domain: it is a second dimerization interface, a ligand binding pocket, and a coregulator binding surface, and activation function 2 (AF2) acts as a binding site for coregulator proteins.

The hPPARγ is mainly expressed in adipose tissue, macrophages, vascular smooth muscle, and in tumors originating from various organs [5]. An early study reported hPPARγ had a key role in regulating adipocyte differentiation [6] and insulin sensitivity [7]. Therefore, modulators of hPPARγ activity have the potential for the treatment of type 2 diabetes. hPPARγ full agonists—glitazone class drugs (TZD) such as pioglitazone and rosiglitazone—are used for the treatment of type 2 diabetes [8]. Despite the beneficial clinical effects of these drugs, TZDs cause several adverse effects, including significant weight gain, peripheral edema, bone loss, and an increased risk of congestive heart failure, mainly arising from their over-activation of hPPARγ [9]. Thus, novel hPPARγ-modulating agents that activate hPPARγ moderately, not fully, clearly need to be developed. Phosphorylation of Ser245 of hPPARγ with cyclin-dependent kinase 5 (Cdk5) is an important post-translational modification of hPPARγ [10]. CDK5 does not alter its adipogenic activity but dysregulated a specific set of genes with important roles in obesity and diabetes. Recently, an hPPARγ antagonist was reported to block hPPARγ phosphorylation and exert anti-diabetic activity with fewer side effects compared with the hPPARγ full agonist by its direct binding to a site near the β-sheet and helix H3 of hPPARγ [11], which was recently shown to be an alternate/allosteric binding site [12].

From a historical point of view, the hPPARγ ligand discovery program was initially focused on the identification of hPPARγ full and partial agonists. Later, molecular pharmacology research indicated the significance of antagonist-mediated hPPARγ signal transduction for drug discovery.

This evidence demonstrates that hPPARγ agonists and antagonists are attractive molecular targets for the treatment of various diseases, including not only type II diabetes and cancer, but also new targets, such as epilepsy [13], allergies [14], and non-alcoholic fatty liver disease (NAFLD) [15]. Therefore, potent hPPARγ-selective ligands should be developed. Previously, we summarized our structural development studies in historical order to create hPPAR subtype-selective agonists [16]. The discovery of many types of hPPAR ligands indicates the validity of our strategy to create subtype-selective hPPAR agonists. In this review, we summarize our structural development studies to create covalent-binding and non-covalent-binding hPPARγ antagonists.

## 2. Working Hypothesis to Create a PPARγ Antagonist from PPARγ Agonist 1

Previously, we reported that nuclear receptor antagonists can be designed and synthesized in accordance with the helix 12 (H12) proper folding inhibition hypothesis [17]. This hypothesis was created based on the results of a number of X-ray crystallographic analyses of NR ligand binding domains (LBDs) with or without a bound agonist, focusing on the agonist-induced proper folding of the H12 of NR LBDs. In the absence of an agonist, H12 exists in various open conformations, which might be favorable for the recruitment of corepressors that induce the transrepression of the NR signal transduction. However, once a full agonist binds to the ligand-binding pocket (LBP) of the NR—forming a hydrogen bond network with the receptor—H12 forms a closed, solvent-exposed active conformation, which consequently stabilizes the AF-2 region of the NR. This stabilization facilitates the dissociation of the corepressor followed by the association of a coactivator with the receptor to induce its transactivation. Thus, an agonist that induces the proper folding of H12 might be required to initiate the transcriptional processes. On the basis of these considerations, ligands that bind to the LBP and interfere with the proper folding of H12 should be antagonists of the corresponding NR. In accordance with this hypothesis, we successfully designed and synthesized a range of NR antagonists, including retinoic acid receptor antagonists [18], liver X receptor antagonists [19], and farnesoid X receptor antagonists [20].

To create structurally new hPPARγ antagonists based on this working hypothesis, we initially focused on our previously created hPPARγ-selective agonist MEKT-21 (EC_50_ = 0.08 ± 0.009 μM, E_max_ = 80 ± 3.1% relative to the hPPARγ pan agonist, TIPP-703, in our assay system) as a template [21] because we solved its X-ray crystal structure complexed with hPPARγ LBD. From the complex structure, we noted that the flexible α-ethyl phenylpropanoic acid moiety formed a hydrogen bond network with the side chain amino acids of Ser289, His323, Tyr327, and Tyr473 of the hPPARγ LBD. Of these, the hydrogen bond interaction between the carboxylic acid moiety of TIPP-703 with Tyr473 on H12 was critically important for the proper folding of the H12 of the hPPARγ and the transactivation of the target genes [22]. Therefore, an effective approach for antagonist design should be the disruption of this key interaction by the replacement of the carboxyl group of MEKT-21 with other functional groups, such as an acylsulfonamide group (plan A in Figure 1) to obtain two important derivatives: a phenyl sulfonyl amino carbonyl derivative (MEKT-75 (**7a**)) and a benzyl sulfonyl amino carbonyl derivative (MEKT-76 (**7b**)).

The synthetic production of MEKT-75 and MEKT-76 is shown in Figure 1. Salicylaldehyde **1** was alkylated with n-propyl iodide to give salicylaldehyde **2**. Compound **2** was amide alkylated with *t*-butyl carbamate followed by acid hydrolysis to afford aminomethylbenzene derivative **3** as a hydrochloric acid salt. Compound **3** was condensed with 4-(2-pyrimidinyl)benzoic acid in the presence of DEPC and resulting product **4** was regioselectively formylated (**5**), followed by Pinnick oxidation to afford benzoic acid **6**. Compound **6** was condensed with benzenesulfonamide (or benzyl sulfonamide) in the presence of EDC and DMAP to afford the target compounds MEKT-75 and MEKT-76.

We evaluated the transactivation activities of MEKT-75 and MEKT-76 (Figure 2 left). MEKT-75 exhibited significant transactivation activity; however, dose-dependency was not seen and it reached about 50% of the maximal activity obtained from the maximal response of the positive control (10 μM pioglitazone (hPPARγ full agonist)) at the concentration of 10 μM. These experimental results clearly indicated that MEKT-75 is an hPPARγ partial agonist [23].

MEKT-76 exhibited lower transactivation activity compared with the positive control; about 40% of the maximal activity obtained from the maximal response of the positive control at the highest concentration of 30 μM. Of note, MEKT-76 dose-dependently inhibited the transactivation activity elicited by 10 μM of the hPPARγ full agonist, pioglitazone (Figure 2 right). These data indicate that MEKT-76 is a far less potent hPPARγ partial agonist than MEKT-75 and is weak enough to exhibit an antagonistic nature when co-treated with an hPPARγ full agonist [24].

To understand the substantial differences between the transactivation/transrepression activities of these two derivatives, we solved their X-ray crystal structures complexed with the hPPARγ LBD at resolutions of 2.1 Å and 2.2 Å, respectively. A crystal of the complex was obtained by soaking a crystal of the homodimer in ligand solution. The results are summarized in Figure 3.

The hPPARγ LBD–MEKT-75 crystal formed a homodimer and each hPPARγ LBD bound to one molecule of MEKT-75 (Figure 3A). It is important to note that each structural fold of hPPARγ LBD was similar except from the end of helix 11 (H11) to the C-terminal H12 region. Two bound ligands in each hPPARγ LBD–MEKT-75 complex had a similar three-dimensional structure and were situated in arm2 and arm3 of the binding pocket of the hPPARγ LBD. One of the structural folds in the homodimer was similar to that obtained from the hPPARγ LBD–rosiglitazone (hPPARγ full-agonist) complex (Figure 3A,B) [25]. Therefore, we tentatively designated the former structure as the fully active form of the PPARγ LBD and the latter structure as a non-fully active form.

In the fully active form, MEKT-75 has a U-shaped structure and the acylsulfonamide group of MEKT-75 is positioned near H11. Interactions of three of the five key amino acids (Ser289, His323, Tyr327, His449, and Tyr473), which are critical for the construction of hydrogen bond networks, were conserved in the one LBD complexed with MEKT-75. This appeared to be sufficient to support the fully active LBD structure. In contrast, in the non-fully active form, another interaction was noted: His266 was located close to the (pyrimidin-2-yl)phenyl moiety of MEKT-75, resulting in a hydrophobic interaction between His266 and the phenyl group and between His266 and the pyrimidin-2-yl group. These additional interactions caused bound MEKT-75 to move to the right, so that the distance from the phenyl sulfonyl amino carbonyl group of MEKT-75 to the side chains of the surrounding amino acids became longer. As a result, the hydrogen bond network was weaker, suggesting that the H12 region is not restricted to the appropriate location for full activity.

The hPPARγ LBD–rosiglitazone complex also formed a homodimer in its crystal form, but each LBD was present in its fully active form (Figure 3B). This result indicated that the full agonist induced a conformational change of the non-fully active hPPARγ LBD to the fully active LBD, presumably by facilitating a tight hydrogen bond network with the LBD.

The hPPARγ LBD–MEKT-76 crystal also formed a homodimer, but only one molecule of MEKT-76 was bound to the hPPARγ LBD. The structural fold of the bound hPPARγ LBD was similar to that of the non-fully active form of the LBD complexed with MEKT-75. The 4-(pyrimidin-2-yl)benzamide side chain of MEKT-76 was docked into a large pocket composed of H2, β2, H3, and H5, which made hydrophobic contact with the surrounding amino acid residues. The benzyl sulfonyl amino carbonyl side chain was docked between the side chains of the amino acids of H3 and H5, which made hydrophobic contact with the surrounding amino acid residues. The *n*-propoxy side chain of MEKT-76 was located between the side chains of the amino acids of H6 and H7 and H11. Only His449—a key amino acid (Ser289, His323, Tyr327, His449, and Tyr473) for hPPARγ-agonistic activity—contributed to the formation of the hydrogen bond interactions. Therefore, the present X-ray structure is consistent with the idea that MEKT-76 is a very weak hPPARγ partial agonist, because both hPPARγ LBDs in the homodimeric complex had a non-fully active structure and MEKT-76 was loosely bound in the binding pocket of only one of the homodimer complexes.

The rank order of the interaction energy between rosiglitazone, MEKT-75, and MEKT-76 correlated with the hPPARγ agonistic nature. Figure 4 shows the binding energy results of rosiglitazone, MEKT-75, and MEKT-76. The total interaction energy of rosiglitazone (hPPARγ full agonist) to the full agonist form of the hPPARγ LBD was −125.24 kcal/mol (calculated using the van der Waals (vdW) interaction energy (−43.68 kcal/mol) and electrostatic (ele) interaction energy (−81.56 kcal/mol); the relative vdW interaction energy (vdW/volume) was −0.14 kcal/mol and the relative ele interaction energy (ele/volume) was −0.25 kcal/mol) [26]. This indicated that rosiglitazone was docked into the full agonist form of hPPARγ LBD primarily with electrostatic interactions. For the MEKT-75 full agonist form of the hPPARγ LBD, the total interaction energy of MEKT-75 to the full agonist form of the hPPARγ LBD was −137.65 kcal/mol (calculated using the vdW interaction energy (−78.95 kcal/mol) and ele interaction energy (−58.70 kcal/mol)—the relative vdW interaction energy (vdW/volume) was −0.17 kcal/mol and the relative ele interaction energy (ele/volume) was −0.13 kcal/mol). These data indicated that MEKT-75 was docked into the full agonist form of the hPPARγ LBD with comparable van der Waals interactions and electrostatic interactions. Although the relative vdW interaction energy was comparable to that of rosiglitazone, the relative ele interaction energy (ele/volume) was only one-half of that of rosiglitazone. This decreased relative ele interaction energy might contribute to the hPPARγ partial agonist nature of MEKT-75.

For both non-full agonist forms of the hPPARγ LBD complexed with MEKT-75 and MEKT-76, the total interaction energy of the non-full agonist form of the hPPARγ LBD was about −114 kcal/mol (calculated using the vdW interaction energy (−75 kcal/mol) and ele interaction energy (−39 kcal/mol); the relative vdW interaction energy (vdW/volume) was −0.16 kcal/mol and the relative ele interaction energy (ele/volume) was −0.08 kcal/mol). These data indicated that both compounds were docked into the non-full agonist form of the hPPARγ LBD primarily with van der Waals interactions.

Although MEKT-76 is only a homolog of MEKT-75, the binding mode of MEKT-76 complexed with the hPPARγ LBD was quite different from that obtained with MEKT-75. The bound parts of the benzyl sulfonyl amino carbonyl side chain and the *n*-propoxy side chain of MEKT-76 were docked into the opposite binding pockets compared with the corresponding side chains of MEKT-75. Figure 5 shows the superimposed structures of bound MEKT-75 and bound MEKT-76 complexed with the hPPARγ LBD, and Figure 5B,E and Figure 5C,F show the binding modes of MEKT-75 and MEKT-76 complexed with the hPPARγ LBD, respectively. In Figure 5D, the hinge region benzene rings poorly overlap with each other. The hinge region benzene ring of MEKT-76 was shifted by a distance equivalent to one benzene ring compared with its position in MEKT-75. The most significant difference between the binding modes of both compounds was the position of the two substituents at the hinge region benzene ring. These substituents of MEKT-75 and MEKT-76 were in opposite directions (180° difference). For MEKT-75 complexed with the hPPARγ LBD, the phenyl sulfonyl amino carbonyl group faced Tyr327, which is a critically important amino acid for full agonistic activity and has hydrogen bonding interactions with the surrounding key amino acids. However, in the MEKT-76 complex, the benzyl sulfonyl amino carbonyl group was flipped in the other direction and the *n*-propoxy group faced Tyr327, but had no apparent hydrogen bond interactions with the surrounding key amino acids.

Recent structural biology studies indicated that the hPPARγ LBP can be divided into two regions. One is for the orthosteric binding pocket and the other is for the alternate (allosteric) binding pocket (Figure 5A) [27]. The orthosteric binding pocket is defined by helices 5, 7, and 11, whereas the alternate binding site consists of helices H2, H3, and β-sheets. The entrance to the ligand binding site of the hPPARγ LBD was reported to be between H3 and the three-stranded antiparallel β-sheets [28].

We thus speculated that the binding of MEKT-76 at the alternate (allosteric) binding pocket shielded the entrance site of the ligands, which might explain why MEKT-76 exhibited antagonistic activity when co-treated with an hPPARγ full agonist.

## 3. Working Hypothesis to Create a PPARγ Antagonist from PPARγ Agonist 2

Although MEKT-76 demonstrated hPPARγ antagonistic activity, it involved intrinsic hPPARγ agonistic activity to some extent. Therefore, a strategy to delete the acidic carboxylic acid functionality of the hPPARγ agonist (plan A) was not effective for developing a complete hPPARγ antagonist.

As described previously, the hydrogen bond interaction with Tyr473 on the H12 of the hPPARγ LBD was considered critical for the proper folding of the H12 of the hPPARγ LBD, which stabilized the water-accessible, closed conformation, fully active form of H12. Therefore, an effective approach for antagonist design must block this key interaction. However, the X-ray crystal structures of MEKT-75 complexed with the hPPARγ LBD indicated that although the acylsulfonamide group of MEKT-75 had no direct hydrogen bond interaction with Tyr473, it formed hydrogen bonds with the other side chain amino acids of Tyr327 and His449 around the H12. Consequently, the overall structural fold of the hPPARγ LBD–MEKT-75 complex—including the H12 position—was very similar to that of the complex obtained with the hPPARγ full agonist, rosiglitazone. A lack of interactions between MEKT-75 and Tyr473 might explain the partial agonist nature of MEKT-75 because it is unable to completely inhibit the proper folding of H12. On the basis of this, we speculated that a folding inhibition-type full antagonist of hPPARγ should have no direct hydrogen bond interactions with the Tyr473 of H12 and should push the H12 out of position. This design concept is illustrated in Figure 6, plan B.

Plan B contains two ideas. One is the removal of the acidic carboxyl group to disable hydrogen bond interactions with Tyr473, as in plan A. This is necessary to inhibit the full hPPARγ agonistic nature of the ligand. The other idea is the rigidification and expansion of the residual conformationally-flexible 3-phenylpropyl group of MEKT-21, which is expected to push the H12 out of the full agonist form position. Accordingly, we focused on a reversed amide linkage as the rigid tether, aiming to compensate for the loss of affinity caused by the deletion of the hydrogen bond interactions with Tyr473 [29].

The procedure to synthesize the reversed amide derivatives is depicted in Figure 2. 5-nitrosalicylaldehyde (**8**) was *n*-propylated and then amide-alkylated with *t*-butylcarbamate in the presence of trifluoroacetic acid and triethylsilane [30], followed by the removal of the *N*-Boc protecting group to afford aminomethyl benzene derivative **10** as a hydrochloric acid salt. Compound **10** was condensed with 4-(2-pyrimydyl)benzoic acid and then the nitro group was reduced to afford aniline **12**. Compound **12** was condensed with the appropriate carboxylic acid to afford the reversed amide derivatives **13a–r**.

As expected, the initial synthesized phenyl (**13a**), benzyl (**13b**), and 2-phenylethyl (**13c**) derivatives exhibited decreased hPPARγ agonistic activity compared with MEKT-21 (Figure 7), which might be related to the deletion of the acidic carboxyl group. Compounds **13a** and **13b** showed hPPARγ partial agonistic activities with moderate potency and low efficacy compared with the lead compound, MEKT-21. The increase in efficacy was in the order of phenyl < benzyl < 2-phenethyl derivatives. These results prompted us to speculate that the substituents bearing single bonds such as the benzyl or 2-phenethyl group did not efficiently push H12 out of position because of its conformational flexibility. Of these compounds, **13a** was the most potent hPPARγ partial agonist with low efficacy; therefore, we focused on **13a** as the next lead compound and prepared **13a** derivatives. The introduction of more rigid and bulkier substituents such as 4-ethynylphenyl (**13d**), 1-naphthyl (**13e**), 2-naphthyl (**13f**), and 4-biphenyl group (**13g**) decreased the efficacy of the derivatives compared with **13a**. The introduction of a styryl group (**13h**) promoted antagonist activity (IC_50_ = 4.7 ± 1.3 μM, I_max_ = 66 ± 2.3%) with no concomitant agonist activity. Finally, compound **13i** (MEKT-160) exhibited the most potent hPPARγ antagonist activity (IC_50_ = 0.17 ± 0.02 μM) in this series. We speculated that the rigid phenyl alkynyl moiety of **13i** might effectively interfere with the proper folding of H12, resulting in robust antagonist activity; however, this was a misunderstanding based on our X-ray crystallographic study as described later. Next, we attempted to modify the distal aromatic ring of **13i** to obtain potent PPARγ antagonists without any concomitant agonist activity. The SAR results indicated that the introduction of substituents at the 2-position of the distal benzene ring tended to diminish the PPARγ agonist activity. A 2-methyl derivative (**13j**) did not exhibit agonist activity although a 3-methyl derivative (**13k**) and 4-methyl derivative (**13l**) retained very low efficacy. Notably, 2-chloro or 2-bromo derivatives (**13m** and **13n**) had full antagonistic activity and blocked hPPARγ activation by 30 μM pioglitazone completely with an IC_50_ value of 0.2–0.4 μM without concomitant hPPARγ agonist activity. A 3-chlorophenyl derivative (**13o**) did not exhibit the same antagonist profile as **13m**. These results suggested that the appropriate steric bulkiness and proper substitution position of the halogen were necessary to obtain a full antagonist profile.

Next, we characterized the ligand-mediated cofactor recruitment elicited by the hPPARγ full agonist rosiglitazone (**1**) and hPPARγ antagonist MEKT-160 (**13i**) using a mammalian two-hybrid assay system [31]. Rosiglitazone (**1**) potently enhanced the positive interaction of the coactivator SMRT with hPPARγ; however, no apparent interaction was detected for the hPPARγ antagonist MEKT-160 (Figure 8 left). Contrary to the coactivator recruitment, the corepressor NCoR–hPPARγ interaction was potently enhanced by MEKT-160 compared with rosiglitazone (Figure 8 right). Thus, the effects of MEKT-160 on cofactor recruitment to hPPARγ were markedly different from those of the hPPARγ agonist.

A previous study reported that PPARγ-corepressor association was involved in the development of obesity and diabetes [32,33]. Thus, compounds that regulate the corepressor recruitment profile might be candidate agents for treating obesity and diabetes or useful biological tools to evaluate PPARγ-corepressor associations in biological models of obesity and diabetes.

hPPARγ is a well-known master regulator of adipogenesis; it promotes the conversion of a variety of preadipocytes and stem cell lines into mature adipocytes [34]. Thus, to characterize the effect of hPPARγ on adipocytes, we investigated the antagonistic activity of **13m** (a 2-chloro derivative of MEKT-160) on rosiglitazone-induced adipocyte differentiation. Preadipocyte 3T3-L1 cells were induced to differentiate by treatment with 1 μM rosiglitazone and 5 μg/mL insulin in the presence or absence of **13m**. As shown in Figure 9, rosiglitazone promoted adipocyte differentiation, as shown by an increase in the lipid content demonstrated by Oil red O staining. However, **13m** significantly inhibited rosiglitazone-induced adipocyte differentiation at 1 μM, and completely suppressed it at 10 μM. This result indicated that **13m** acts as a hPPARγ full antagonist at the cellular level.

## 4. Arylalkynyl Amide-Type hPPARγ Antagonists Bind Covalently to hPPARγ via a Unique Binding Mode

Our structural development studies succeeded in creating the arylalkynyl amide-type hPPARγ antagonists such as MEKT-160 starting from the hPPARγ partial agonist, MEKT-21. We expected that the steric bulkiness and rigid arylalkynyl amide moiety of the compounds would interfere with the hPPARγ H12 and push the H12 out of the full agonist form position. To confirm this, an X-ray crystallographic structure analysis of the representative arylalkynyl amide-type hPPARγ antagonist complexed with the hPPARγ LBD could be performed. However, prior to that, we checked the chemical reactivity of arylalkynyl amides.

Arylalkynyl amides and esters are Michael acceptor structures, and various nucleophiles, such as sulfhydryl groups and amino groups, react with an alkynyl amide via Michael addition to form a nucleophile-substituted alkenyl amide (Michael adduct) [35,36]. Previously, oxidized unsaturated fatty acids and their derivatives, such as 4-oxodocosahexaenoic acid (4-oxoDHA), 6-oxooctadecatrienoic acid (6-oxo-OTE), and 15-deoxy-delta-12,14-prostaglandin J2 (15-dPGJ2), were reported to bind covalently to the Cys285 of the hPPARγ LBD [37,38]. In addition, the synthetic hPPARγ antagonists, GW9662 and T0070907 were shown to bind covalently to the Cys285 of the hPPARγ LBD [39,40]. These derivatives all contain a nucleophilic substitution spot. Considering this, we speculated that alkynyl amide-type hPPARγ-selective antagonists, such as MMT-160, might also bind covalently to the hPPARγ LBD, via the Cys285 in the H3 of the hPPARγ LBD. To confirm this, electrospray ionization-mass spectrometry (ESI-MS) evaluation of the hPPARγ LBD after incubation with various ligands was performed (Figure 10) [41]. The apo form of the hPPARγ LBD had a molecular mass of 31,411 by ESI-MS, and in the presence of rosiglitazone or MEKT-21 (negative controls), the mass signal did not change. Conversely, in the presence of the covalent binder GW-9662, the mass signals shifted upward to a molecular mass that corresponded to the apo hPPARγ LBD + GW-9662 derivative. This clearly indicated that ESI-MS is useful to evaluate covalent binding. For MMT-160 and its 4-piperidine derivative, both of which contain an alkynyl amide structure, the mass signals were shifted upward to a higher molecular weight, with the molecular masses corresponding to the apo hPPARγ LBD + MMT-160 and apo hPPARγ LBD + 4-piperidine derivatives, respectively. Of note, a saturated carbonyl derivative of MMT-160 did not affect the mass signal of the apo form of the hPPARγ LBD.

The ESI-MS analysis of the hPPARγ LBD after incubation with MMT-160 and its 4-piperidine derivative clearly indicated that these alkynyl amide derivatives bound covalently to the hPPARγ LBD.

To investigate the precise covalent binding mode of alkynyl arylamide-type hPPARγ antagonists further, we solved the X-ray crystallographic structure of MMT-160 complexed with the hPPARγ LBD homodimer at 3.0 Å resolution. A binary complex was obtained by co-crystallization (protein data bank accession code: 7WOX) (Figure 11A–E).

Only one hPPARγ LBD in the homodimer (designated as chain A) was able to be refined in the structure of MMT-160 (Figure 11A). The phenyl acetylenyl (Michael acceptor) portion of MMT-160 was docked in the Y1 arm of the binding pocket extending through the space between the helices, H3, H4, and H12, and formed hydrophobic interactions (Figure 11B). Distinct electron density was observed around the sulfhydryl group of Cys285 and the distance between the sulfhydryl group of Cys285 and the benzylic carbon of MMT-160 was estimated as 1.80 Å, consistent with the theoretical C–S bond length (Figure 11B). No apparent hydrogen bond interactions were observed between MMT-160 and five critical amino acids: Ser289, His323, Tyr327, His449, and especially Tyr473. The lack of these interactions might explain why MMT-160 did not exhibit obvious hPPARγ agonistic activity (Figure 11B).

Surprisingly, the two other hydrophobic side chains of MMT-160 were found to dock into the unexpected binding sites. The *n*-propoxy side chain hosted in the Y2 arm, situated between H3 and the β-sheet, also formed hydrophobic interactions (Figure 11C). What was most surprising was that the pyrimidine-2-yl benzamide of MMT-160 was not buried in the large binding pocket of the hPPARγ LBD but extended out from the pocket and the carbonyl oxygen formed hydrogen bond interactions with Gly284 (Figure 11D). These data indicated that MMT-160 was docked into to the hPPARγ LBD—mainly via hydrophobic interactions with the Y1- and Y2-arm amino acids—in addition to its covalent binding to CYS285.

As described previously, the entrance to the ligand binding site of the hPPARγ LBD was reported to be between H3 and a three-stranded antiparallel β-sheet. We focused our attention on the entrance site. Figure 11F–H clearly shows that MMT-160 covalently bound to CYS285 on the H3 shielded the entrance site of the ligand more effectively than GW-9662.

We found that the arylalkynyl amide-type hPPARγ antagonists bound covalently to the CYS285 of the PPARγ LBD via Michael addition. The X-ray crystallographic analysis of the hPPARγ LBD complexed with MMT-160 indicated its unique binding mode, which is quite different from the recently reported PPARγ antagonists of SR-10171 (phenoxyacetic acid derivative) and SR-11023 (phenylacetic acid derivative) [42], thus supporting some of the antagonistic nature of MMT-160.

## 5. Conclusions

Two types of hPPARγ antagonists: a non-covalently bound antagonist, MEKT-76, and a covalently bound antagonist, MMT-160, were created. Generally, when the allosteric binding site occupancy ratio with the ligand increased, the ligand tended to show a greater antagonistic effect (Figure 12).

Both series of compounds were structurally derived from our original hPPARγ agonist, MEKT-21.

Our study demonstrated that the agonist–antagonist switching concept—a simple medicinal chemistry strategy for the design of nuclear receptor antagonists—can be used to develop hPPARγ antagonists. This strategy is also applicable to create other nuclear receptor antagonists [43,44]. However, we speculated that our strategy created folding inhibition-type antagonists of hPPARγ, which have no direct hydrogen bond interactions with the Tyr473 of H12, resulting in pushing the H12 out of position. Is our speculation correct? The answer is “not yet!”

A recent X-ray crystallographic study of the hPPARγ ligands SR-10171 (inverse agonist) and SR-11023 (antagonist) complexed with the hPPARγ LBD confirmed our speculation/working hypothesis. These compounds are thought to be H12 folding inhibition-type antagonists of hPPARγ. Both ligands are docked between the H3 and β-sheet of the hPPARγ LBD and wrap around the solvent-exposed face of H3. Importantly, the hydrophobic tails of SR10171 and SR11023 form hydrophobic interactions between the H3 and H12 of the hPPARγ LBD and stabilize H12 at H3, away from the AF2 coactivator-binding surface.

On the basis of this, our next idea (plan C) involves the introduction/replacement of other hydrophobic substituents at the proper position of MMT-160, which will interact hydrophobically with the H12 of the hPPARγ LBD and pull the H12 to a position near to the newly introduced/replaced hydrophobic substituents. We expect that the hot spot should be an *n*-propoxy group and/or 4-(pyrimidin-2-yl)benzamide moiety of MMT-160. We are now conducting further structural studies related to the development of covalent-binding hPPARγ ligands.

## Data Availability

Data sharing not applicable.

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
