# Peer review of "Structural Biology Inspired Development of a Series of Human Peroxisome Proliferator-Activated Receptor Gamma (PPARγ) Ligands: From Agonist to Antagonist"

_ijms, 2023, doi:10.3390/ijms24043940_

Round 1

Reviewer 1 Report

This review about structural biology and development of a series of human PPARgamma ligands is of interest. However, in my opinion, it could be suggested in which diseases they could find an application. For instance, the authors report that “hPPARγ agonists and antagonists are attractive molecular targets for the treatment of various diseases.” Could you please discuss in which diseases they could find an application? For instance, it was recently published that the peroxisome proliferator-activated receptor gamma could be a putative target for epilepsy treatment (Senn et al., 2023). Moreover, it was found that PPARγ might be a useful marker of response to the anticonvulsant EP-80317 (Lucchi et al., 2017). Finally, the antiseizure effects of cannabidiol were associated with upregulation of PPARγ in the hippocampal CA3 region (Costa et al., 2022).

Line 60 “…fewer side effects compared with [xxxx] by its direct…” Please check!

Author Response

*Thank you reviewer1 for the positive comment.
Considering to the reviewer1's comment, the author adds some sentences in the revised TEXT, and add the corresponding refs. 

*Line 60 “…fewer side effects compared with [xxxx] by its direct…” Please check!
>>>>> The author correctly revised in the revised TEXT.

Reviewer 2 Report

 The manuscript deals with peroxisome proliferator-activated receptor gamma which is a target for several diseases including cancer.

The authors brief their research realted to covalent and non-covalent antagonist.

The manuscript is well written and pertinent, I have no major issues to report other than minor editorial changes that can be made during the final typesetting

Author Response

*The author thanks the reviewer2 for the positive comments. The reviewer carefully re-checked the TEXT, and corrected in the revised TEXT.

Reviewer 3 Report

As author has submitted the work under "Review" category, 

but authors stated in conclusion section - " We succeeded in creating two types of hPPARgamma antagonists: a non-covalently bound antagonist, MEKT-76, and a covalently bound antagonist, MMT-160. Generally, when the allosteric binding site occupancy ratio with the ligand increased, the ligand tended to show a greater antagonistic effect." this is bit confusing.... Please clarify it.

Working hypothesis to create a PPARgamma antagonist from PPARgamma agonist 1 - In this section the citations are majorly coming from the authors published data, the authors make check with other authors and research group data also...

Similar is observe in other two hypothesis...

The authors should check with diverse dataset.

Author Response

*The author thanks the reviewer3 for the positive comment. Also the author thanks the reviewer3 for pointing out the check points.  Based on the reviewer3, the author corrected the expression of some sentences in the revised TEXT.

*The author adds some other researcher's report concerning about agonist-antagonist exchange as ref 43 and ref 44 in the revised TEXT.

Round 2

Reviewer 3 Report

As the authors have added a few supporting articles in the reference and made minor corrections in the text, that is acceptable.

Minor language correction is needed please check it.

Author Response

Thank you reviewer3 for the proper suggestion.

We re-check the language with a native English-speaking colleague.